# Seroepidemiology of pertussis in Huzhou: A population-based, cross-sectional study

Yan Liu ⊕, Chao Zhang⊕, Yuda Wang, Xiaofu Luo, Guangtao Liu, Zizhe Zhang, Jianyong Shen*

Huzhou Center for Disease Control and Prevention, Huzhou, China

⊕ These authors contributed equally to this work.
* 32288746@qq.com

## Abstract

### Purpose

The resurgence of pertussis has occurred around the world. However, the epidemiological profiles of pertussis cannot be well understood by current diseases surveillance. This study was designed to understand the seroepidemiological characteristics of pertussis infection in the general population of Huzhou City, evaluate the prevalence infection of pertussis in the population, and offer insights to inform adjustments in pertussis prevention and control strategies.

### Methods

From September to October 2023, a cross-sectional serosurvey was conducted in Huzhou City, involving 1015 permanent residents. Serum samples were collected from the study subjects, and pertussis toxin IgG antibodies (Anti-PT-IgG) were quantitatively measured using enzyme-linked immunosorbent assay (ELISA). The analysis included the geometric mean concentration (GMC) of Anti-PT-IgG, rates of GMC≥40IU/mL, ≥100IU/mL, and <5IU/mL. Stratified comparisons were made based on age, vaccination history, and human categories.

### Results

Among the 1015 surveyed individuals, the geometric mean concentration (GMC) of Anti-PT-IgG was 10.52 (95% CI: 9.96–11.11) IU/mL, with a recent infection rate of 1.58%, a serum positivity rate of 11.43%, and a proportion with <5IU/mL of 40.49%. Among 357 children with clear vaccination history, susceptibility decreased with an increasing number of vaccine doses (Z = -6.793, $P$ < 0.001). The concentration of Anti-PT-IgG exhibited a significant post-vaccination decline over time (Z = -5.143, $P$ < 0.001). In women of childbearing age, the GMC of Anti-PT-IgG was 7.71 (95% CI: 6.90–8.62) IU/mL, with no significant difference in susceptibility among different age groups ($\chi^2$ = 0.545, $P$ = 0.909). The annual pertussis infection rate in individuals aged ≥3 years was 9321 (95%CI: 3336–16039) per 100,000, with peak infection rates in the 20–29, 40–49, and 5–9 age groups at 34363 (95%CI: 6327–

**Data Availability Statement:** All relevant data are within the manuscript and its Supporting Information files.

**Funding:** The fundings information for this study is as follows: Applied Research Project of Public

Welfare of Huzhou Bureau of Science and Technology(2022GYB10), Huzhou Center for Disease Control and Prevention key discipline: acute Infectious Diseases(ZDXK202202). The funders had no role in study design, data collection and analysis decision to publish or preparation of the manuscript.

**Competing interests:** The authors have declared that no competing interests exist.

66918) per 100,000, 22307.72 (95%CI: 1380–47442) per 100,000, and 18020(95%CI: 1093–37266) per 100,000, respectively.

## Conclusions

In 2023, the actual pertussis infection rate in the population of Huzhou City was relatively high. Vaccine-induced antibodies exhibit a rapid decay, and the estimated serum infection rate increases rapidly from post-school age, peaking in the 20–29 age group. It is recommended to enhance pertussis monitoring in adolescents and adults and refine vaccine immunization strategies.

## Introduction

Pertussis, caused by *Bordetella pertussis*, is a severe acute respiratory infectious disease [1]. Despite the availability of vaccines for pertussis prevention, it remains a globally significant cause of morbidity and mortality, especially among children aged 5 and below. The World Health Organization (WHO) estimated over 169,000 reported cases of pertussis worldwide in 2018, with the majority originating from developing countries. Current research indicates a sustained high level of infection in the population [2–4], and there is a consensus that the burden of pertussis may be seriously underestimated [5]. Pertussis continues to be a significant public health concern, with a noticeable increase in reported cases in China in recent years [6], The annual reported incidence has risen from 0.12 per 100,000 in 2009 to 2.15 per 100,000 in 2019 [7], accompanied by outbreaks [8]. Even in countries with high vaccine coverage, pertussis resurgence has been observed [9], with a notable increase in cases among infants (age <1 year), adolescents (10-19years), and adults (≥ 20 year), a phenomenon internationally termed "pertussis resurgence"[10].

Studies suggest that the protective efficacy of pertussis vaccines may only last 4–12 years [11], leading to the reporting of cases in older children, adolescents, and adults. Surveillance data from Huzhou City in 2022 shows a significant increase in pertussis incidence.[12]. Pertussis toxin (PT) is considered a specific protective antigen for it is an important component of acellular pertussis vaccines, and Anti-PT-IgG is a specific antibody against B. pertussis [13]. Elevated Anti-PT-IgG in the serum of healthy individuals is considered a sensitive indicator of recent *B. pertussis* infection [4], making serological monitoring essential for identifying mild and asymptomatic cases and better assessing population infection levels [14].

Current serological studies on pertussis often focus on hospital case populations [15] or specific groups [16]. with significant variations in seroprevalence across different regions [7,17–19]. This study, conducted in Huzhou, China, represents a cross-sectional survey of pertussis toxin antibody levels in a healthy population. The aim is to understand the pertussis infection levels and susceptibility in the entire population of Huzhou, providing a basis for refining pertussis prevention strategies.

## Materials and methods

### Survey subjects and serum samples

Our cross-sectional study was carried out between September 1st to October 1st 2023. Huzhou city comprises three counties and two districts. A multistage stratified random sampling method was employed to select the participants. One county(Deqing County) and one district (Wuxing District) were selected randomly. After that, randomly select a street within the

chosen county/district. Then the study subjects were randomly selected from the community health examination population within September 1st to October 1st 2023. The participants were sampled across nine age groups: 0–3 months, 4–23 months, 3–4 years, 5–9 years, 10–19 years, 20–29 years, 30–39 years, 40–49 years, and ≥50 years. Exclusion criteria included individuals with respiratory symptoms such as fever, cough, and wheezing in the past two weeks, as well as those with acute or chronic infectious diseases, recent trauma or surgery, malignant tumors, immunodeficiency diseases, liver/kidney dysfunction, and neurological disorders that might impact the results. Based on a Zhejiang province serum Anti-PT-IgG positivity rate of 33.32% [2], with $\alpha = 0.05$, $1-\beta = 0.90$, and d = 0.1p, the minimum required sample size was calculated using the formula $N = u_a^2 p(1-p)/d^2$. The calculated minimum sample size was 769 individuals. Accounting for a 10% loss to follow-up, a minimum of 854 individuals needed to be surveyed. Demographic information such as birthday, gender, age and vaccine history and other relevant factors, along with 2–3 mL of serum specimens, were collected from the study subjects. Vaccine administration details were verified through the Zhejiang Province SaaS-based Cloud Vaccination Prevention Platform. According to the vaccine immunization system database, the pertussis vaccines administered to our study participants include both the DTaP (Diphtheria, Tetanus, acellular Pertussis) vaccine and the pentavalent vaccine.

## Laboratory testing

All samples were stored at -80°C before testing. The enzyme-linked immunosorbent assay (ELISA) test kit (produced by EUROIMMUN Medical Laboratory Diagnostics AG) was used to measure the concentration of Anti-PT-IgG for all the samples. The experimental tests were conducted by Hangzhou Edicon Medical Laboratory Co. Ltd. The test procedure strictly followed the instructions provided with the kit. According to the kit instructions, individuals with an Anti-PT-IgG concentration ≥100 IU/mL, measured more than 1 year after immunization, were considered to have acute infection. A concentration of Anti-PT-IgG ≥40 IU/mL was indicative of serum positivity, suggesting infection within the past year. Individuals with an Anti-PT-IgG concentration <5 IU/mL were classified as susceptible.

## Statistical analysis

The Anti-PT-IgG concentrations were logarithmically transformed, and the geometric mean concentration (GMC) was calculated. Values below the detection limit were included in the statistical analysis using the kit's instructions, with a value of 5 IU/mL assigned for samples registering below the detection threshold. The analysis involved the calculation of the GMC, recent infection rate, serum positivity rate, and susceptibility rate of Anti-PT-IgG across different genders, age groups, immunization histories, and various post-vaccination time intervals, as well as different age groups among women of childbearing age. Single-factor analysis of variance was employed to compare the inter-group differences in GMC levels. The post-vaccination time intervals were categorized into four groups based on quartile ranges: <0.7 years, 0.7–1.47 years, 1.47 years-2.453 years, and 2.43 years-5.59 years. Subgroup analysis for vaccination and women of childbearing age aimed to explore the relationship between vaccine administration, maternal age, and Anti-PT-IgG levels. Inter-group rate comparisons were conducted using the chi-square test or fisher test, and trend analyses utilized the Cochran-Armitage trend test. All statistical tests were two-sided with a significance level ($\alpha$) set at 0.05. After pertussis infection, the average time for the Anti-PT-IgG antibody concentration to decrease below 100 IU/mL was 58.6 days (95% CI: 54.2~63.2) [4]. The cutoff level for recent pertussis infection is set at ≥100 IU/mL. The formula for estimating the annual pertussis infection rate was calculated as follows: 365.25/antibody decay time * (proportion of study subjects with Anti-PT-IgG

≥100 IU/mL) [4]. In order to estimate the infection rate, a bootstrap method was employed based on the interval of antibody decay and the proportion of study subjects with Anti-PT-IgG ≥100 IU/mL. To eliminate the impact of vaccination interference, this study calculated the pertussis infection rate among individuals aged ≥3 years. Statistical analysis was conducted using R 4.2.2 software.

## Ethics statement

The study was approved by the human research ethics committee of Huzhou Center for Disease Control and Prevention (Ethics Approval Number: HZ2022004). All participants or their guardians were informed and provided with a written informed consent form.

## Results

### Basic characteristic of study population

This study included a total of 1,015 eligible participants, comprising 519 males and 496 females, resulting in a male-to-female ratio of 1.05:1. The gender difference was not statistically significant ($\chi^2 = 0.521$, $P = 0.470$). The age range was 0–83 years, with a median age of 9 years. The distribution among age groups was as follows: 97 individuals aged 0–3 months, 204 aged 4–23 months, 84 aged 3–4 years, 128 aged 5–9 years, 102 aged 10–19 years, 100 aged 20–29 years, 100 aged 30–39 years, 100 aged 40–49 years, and 100 aged ≥50 years.

### Prevalence of Anti-PT-lgG

There were 116 individuals were seropositive among the 1015 subjects. The mean seropositivity of Anti-PT-IgG was 11.43%(95%CI:9.47%-13.39%), with a GMC of 10.52(95%confidence interval[CI]: 9.96–11.11) IU/mL. Among males, 71 individuals were positive, with a positivity rate of 13.68% (95% CI: 10.72%-16.64%), while among females, 45 individuals were positive, with a positivity rate of 9.07% (95% CI: 6.54%-11.95%). The gender difference in positivity rates was statistically significant ($\chi^2 = 5.319$, $P = 0.021$). The GMC for males was 11.34 (95% CI: 10.48–12.28) IU/mL, and for females, it was 9.72 (9.02–10.47) IU/mL, with a statistically significant difference (F = 7.778, $P = 0.005$). There were statistically significant differences in both the Anti-PT-IgG seropositivity rates and GMC levels across different age groups ($\chi^2 = 125.589$, $P < 0.001$; F = 43.422, $P < 0.001$). A total of 411 samples had Anti-PT-IgG <5 IU/mL, indicating a susceptibility rate of 40.49%. The susceptibility rate in the 4–23 months age group was the lowest, with 18 cases, accounting for 8.82%, which was significantly lower than other age groups ($\chi^2 = 145.323$, $P < 0.001$). See Table 1 for details. Among 18 cases, one had no vaccination history, eight received one dose, four received two doses, three received three doses, and two received four doses.

### The relationship between pertussis vaccine doses and Anti-PT-IgG

We queried the vaccination records of 413 infants and young children through the Zhejiang Provincial Vaccine Administration System. Among them, 56 individuals had an unknown vaccination history, leaving 357 infants and young children with a clear vaccination history for analysis. The number of individuals who received 0, 1, 2, 3, and 4 doses were 54, 34, 16, 71, and 182, respectively. Among the 54 unvaccinated individuals, 50 belong to the 0–3 months category. The GMC of Anti-PT-IgG for each group was 9.18 (95% CI: 7.45–11.33) IU/mL, 6.44 (95% CI: 5.37–7.72) IU/mL, 14.15 (95% CI: 8.21–24.38) IU/mL, 25.27 (95% CI: 20.36–31.35) IU/mL, and 17.57 (95% CI: 15.37–20.09) IU/mL, respectively. The differences were statistically significant (F = 20.193, $P < 0.001$). The GMC for individuals receiving 3 or 4 doses

**Table 1. Distribution of Anti-PT-IgG levels in the general population of Huzhou City in 2023.**

| groups | Number | GMC IU/mL (95%CI) | F | P | Anti-PT-lgG<5IU/mL N(%) | $\chi^2$ | P | Anti-PT-lgG≥40IU/mL N(%) | $\chi^2$ | P | Anti-PT-lgG≥100IU/mL) N(%) | $\chi^2$ | P |
|---|---|---|---|---|---|---|---|---|---|---|---|---|---|
| gender | | | 7.778 | 0.005 | | 6.005 | 0.014 | | 5.319 | 0.021 | | 2.019 | 0.155 |
| male | 519 | 11.34(10.48–12.28) | | | 191(36.80) | | | 71(13.68) | | | 11(2.12) | | |
| female | 496 | 9.72(9.02–10.47) | | | 220(44.35) | | | 45(9.07) | | | 5(1.01) | | |
| age | | | 43.422 | <0.001 | | 145.323 | <0.001 | | 125.589 | <0.001 | | - | <0.001[a] |
| 0-3M | 97 | 7.83(6.86–8.93) | | | 55(56.70) | | | 2(2.06) | | | 0(0.00) | | |
| 4-23M | 204 | 24.83(21.98–28.06) | | | 18(8.82) | | | 67(32.84) | | | 7(3.43) | | |
| 3-4Y | 84 | 11.87(9.91–14.22) | | | 22(26.19) | | | 10(11.90) | | | 0(0.00) | | |
| 5-9Y | 128 | 8.31(7.29–9.47) | | | 65(50.78) | | | 7(5.47) | | | 2(1.56) | | |
| 10-19Y | 102 | 8.00(6.95–9.20) | | | 54(52.94) | | | 7(6.86) | | | 0(0.00) | | |
| 20-29Y | 100 | 7.79(6.56–9.25) | | | 67(67.00) | | | 9(9.00) | | | 4(4.00) | | |
| 30-39Y | 100 | 7.08(6.50–7.71) | | | 45(45.00) | | | 0(0.00) | | | 0(0.00) | | |
| 40-49Y | 100 | 8.70(7.50–10.10) | | | 44(44.00) | | | 5(5.00) | | | 2(2.00) | | |
| ≥50Y | 100 | 9.49(8.10–11.12) | | | 41(41.00) | | | 9(9.00) | | | 1(1.00) | | |
| Total | 1015 | 10.52(9.96–11.11) | | | 411(40.49) | | | 116(11.43) | | | 16(1.58) | | |

significantly higher than those receiving ≤2 doses. There was a significant difference in sero-positivity rates among individuals receiving different doses ($\chi^2$ = 24.517, $P$ < 0.001). As the number of vaccine doses increased, the seropositivity rate of Anti-PT-IgG significantly increased (Cochran-Armitage Trend Test Z = 3.941, $P$ < 0.001). Similarly, with an increase in the number of vaccine doses, susceptibility decreased (Cochran-Armitage Trend Test Z = -6.793, $P$ < 0.001). Refer to Table 2 for details.

For the 182 individuals who completed the full 4 doses of pertussis vaccination, an analysis was conducted on the time elapsed after the last vaccination and Anti-PT-IgG concentrations. The post-vaccination time was divided into four groups based on quartiles: <0.7 years, 0.7–1.47 years, 1.47–2.43 years, and 2.43–5.59 years. The GMC of Anti-PT-IgG for each group was 31.56 (95% CI: 23.45–42.47) IU/mL, 27.04 (95% CI: 21.59–33.87) IU/mL, 15.64 (95% CI: 12.62–19.40) IU/mL, and 9.04 (95% CI: 7.35–11.12) IU/mL, respectively. The concentration differences among the four groups were statistically significant (F = 24.077, $P$ < 0.001). With an increase in the time elapsed after vaccination, Anti-PT-IgG concentrations exhibited a clear declining trend. There were significant differences in the seropositivity rates among the four groups ($\chi^2$ = 27.563, $P$ < 0.001), showing a significant decreasing trend with the longer post-vaccination time (Cochran-Armitage Trend Test Z = -5.143, $P$ < 0.001). Refer to Table 3 for details.

**Table 2. Distribution of Anti-PT-IgG levels in 357 infants and young children with clearly defined vaccination history.**

| group | Number | GMC 95%CI | F | P | Anti-PT-lgG<5IU/mL N(%) | $\chi^2$ | P | Anti-PT-lgG> = 40IU/mL N(%) | $\chi^2$ | P | Anti-PT-lgG≥100IU/mL) N(%) | $\chi^2$ | P |
|---|---|---|---|---|---|---|---|---|---|---|---|---|---|
| Vaccination history | | | 20.193 | <0.001 | | 81.043 | <0.001 | | 24.517 | <0.001 | | - | <0.001[a] |
| 0 | 54 | 9.18(7.45–11.33) | | | 24(44.44) | | | 3(5.56) | | | 1(1.85) | | |
| 1 | 34 | 6.44(5.37–7.72) | | | 26(76.47) | | | 0(0.00) | | | 0(0.00) | | |
| 2 | 16 | 14.15(8.21–24.38) | | | 6(37.50) | | | 2(12.50) | | | 0(.0.00) | | |
| 3 | 71 | 25.27(20.36–31.35) | | | 4(5.63) | | | 23(32.39) | | | 5(7.04) | | |
| 4 | 182 | 17.57(15.37–20.09) | | | 30(16.48) | | | 43(23.63) | | | 1(0.55) | | |
| Total | 357 | 15.41 (13.95–17.03) | | | 90(25.21) | | | 71 (19.89) | | | 7(1.96) | | |

**Table 3. Changes in PT-IgG levels after completed the full 4 doses of pertussis vaccination.**

| group | | GMC | | | | Anti-PT-lgG<5IU/mL | | | Anti-PT-lgG≥40IU/mL | | | Anti-PT-lgG≥100IU/mL) | | |
|---|---|---|---|---|---|---|---|---|---|---|---|---|---|---|
| | Number | 95%CI | F | P | | N(%) | $\chi^2$ | P | N(%) | $\chi^2$ | P | N(%) | $\chi^2$ | P |
| Post-vaccination (Y) | | | 24.077 | <0.001 | | | 23.263 | <0.001 | | 27.563 | <0.001 | | - | 0.209 |
| <0.7 | 38 | 31.56 (23.45–42.47) | | | | 3(7.89) | | | 18 (47.37) | | | 1(2.63) | | |
| 0.7–1.47 | 41 | 27.04 (21.59–33.87) | | | | 0(0.00) | | | 15 (36.59) | | | 0(0.00) | | |
| 1.47–2.43 | 52 | 15.64 (12.62–19.40) | | | | 9(17.31) | | | 7 (13.46) | | | 0(0.00) | | |
| 2.43–5.59 | 51 | 9.04(7.35–11.12) | | | | 18(35.29) | | | 3 (5.88) | | | 0(0.00) | | |
| Total | 182 | 17.57(15.37–20.09) | | | | 30(16.48) | | | 43(23.63) | | | 1(0.55) | | |

## PT-IgG concentration levels in women of childbearing age

In this study, there were a total of 164 women of childbearing age (15–49 years old), and the GMC of Anti-PT-IgG was 7.71 (95% CI: 6.90–8.62) IU/mL. The differences in Anti-PT-IgG concentrations among different age groups were not statistically significant (F = 2.194, P = 0.091). The seropositivity rate for women of childbearing age was 3.05%, and there was a statistically significant difference among different age groups (Fisher P = 0.002). The proportion of women with serum Anti-PT-IgG <5 IU/mL was 54.88%, and the differences in the proportion of <5 IU/mL among different age groups were not statistically significant ($\chi^2$ = 0.545, P = 0.909). The proportion of women aged 20–29 with Anti-PT-IgG ≥100 IU/mL was 8.57%, significantly higher than other age groups (Fisher P = 0.012). Refer to Table 4 for details.

## Pertussis infection rate estimation

The estimated pertussis infection rate was based on 714 individuals aged ≥3 years. The rate of Anti-PT-IgG ≥100 IU/mL in individuals aged ≥3 years was 1.26% (95% CI: 0.58–2.38%). The estimated pertussis infection rate in the population for one year aged ≥3 years in Huzhou City was 9,321 (95% CI: 3,336–16,039) per 100,000. Among them, the estimated infection rate was highest in the 20–29 age group, at 34,363 (95% CI: 6,327–66,918) per 100,000. The second-highest estimated infection rate was in the 40–49 age group, at 22,308 (95% CI: 1,380–47,442) per 100,000. The 5–9 age group had the third-highest estimated infection rate, at 18,020 (95% CI: 1,093–37,266) per 100,000. Refer to Table 5 for details.

## Discussion

Serological monitoring is a crucial research method for understanding the epidemiological characteristics of vaccine-preventable diseases, as it can identify potential infection in populations. Pertussis vaccine is classified as a Class I vaccine in China's immunization program and

**Table 4. Distribution of PT-IgG concentrations in women of childbearing age by age group.**

| group | | GMC | | | Anti-PT-lgG<5IU/mL | | | Anti-PT-lgG≥40IU/mL | | | Anti-PT-lgG≥100IU/mL) | | |
|---|---|---|---|---|---|---|---|---|---|---|---|---|---|
| | Number | 95%CI | F | P | N(%) | $\chi^2$ | P | N(%) | $\chi^2$ | P | N(%) | $\chi^2$ | P |
| Age(year) | | | 2.194 | 0.091 | | 0.545 | 0.909 | | - | 0.002[a] | | - | 0.012 |
| 15–19 | 24 | 8.15 (5.90–11.26) | | | 13(54.17) | | | 2 (8.33) | | | 0(0.00) | | |
| 20–29 | 35 | 9.86 (6.75–14.42) | | | 21(60.00) | | | 3 (8.57) | | | 3(8.57) | | |
| 30–39 | 50 | 6.67 (5.96–7.47) | | | 26(52.00) | | | 0 (0.00) | | | 0(0.00) | | |
| 40–49 | 55 | 7.34 (6.25–8.63) | | | 30(54.55) | | | 0 (0.00) | | | 1(1.82) | | |
| Total | 164 | 7.71(6.90–8.62) | | | 90(54.88) | | | 5 (3.05) | | | 4(2.44) | | |

**Table 5. Pertussis infection level assessment in the population aged ≥3 years in Huzhou City, 2023.**

| Agegroup(year) | Number | PT-Ig≥100IU/mL | PT-Ig≥100IU/mL(%,95%CI) | Infection Rate (/10⁵,95%CI) |
|---|---|---|---|---|
| 3–4 | 84 | 0 | 0.00(0.00–4.30) | 12925(38–28977) |
| 5–9 | 128 | 2 | 1.56(0.19–5.53) | 18020(1093–37266) |
| 10–19 | 102 | 0 | 0.00(0.00–3.55) | 11309(14–23923) |
| 20–29 | 100 | 4 | 4.00(1.10–9.93) | 34363(6327–66918) |
| 30–39 | 100 | 0 | 0.00(0.00–3.62) | 11309(97–24395) |
| 40–49 | 100 | 2 | 2.00(0.24–7.04) | 22308(1380–47442) |
| ≥50 | 100 | 1 | 1.00(0.03–5.45) | 17212 (173–36727) |
| Total | 714 | 9 | 1.26(0.58–2.38) | 9321(3336–16039) |

has significantly reduced the burden of pertussis disease since its introduction [20]. However, in recent years, there has been an upward trend in pertussis incidence despite high vaccination coverage [6,7], Even outbreaks of pertussis have occurred [21]. Therefore, conducting extensive epidemiological studies on pertussis immunity is essential to understand the current infection trends of pertussis.

Currently, there is no unified international standard for the single-serum diagnosis of pertussis, and different countries and regions use different threshold criteria [22]. In this study, the cutoff values recommended by the test kit were employed: ≥100 IU/mL indicating recent infection, ≥40 IU/mL indicating serum positivity, and <5 IU/mL indicating susceptibility to pertussis. The results revealed a seropositivity rate of 11.43% in the entire population of Huzhou City, with a 95% CI of 9.47%-13.39%. This rate was slightly higher than those reported by Zhang et al. (6.7%) in Beijing [23], Wang et al. (6.60%) in a multicenter study across China [13], Chen et al. (5.1%) in Beijing [24], and Zhao et al. (5.14%) in Ningbo [7]. However, it was lower than the rates reported by He et al. (33.32%) in Zhejiang [2], health workers in Spain (31.2%) [16], similar to Chen et al. (11.40%) in Jiangsu [17], and Zhang et al. (8.91%) in Guangdong [25]. The variations in results across different studies are likely attributed to differences in study populations, detection methods, cutoff values, regional disparities [13]. Nevertheless, these findings collectively underscore the widespread existence of pertussis infection in diverse populations [13,26], emphasizing the urgent need to reevaluate the epidemiology of this vaccine-preventable disease and assess its burden.

PT is a protein-based $AB_5$-type exotoxin produced by the bacterium *Bordetella pertussis*, which clearly plays a central role in the process of pertussis infection, while Anti-PT-lgG is a specific antibody against *Bordetella pertussis* [13]. Elevated serum Anti-PT-lgG in healthy individuals who have not been vaccinated for over a year is considered a sensitive indicator of recent pertussis infection [4], and serological monitoring can detect mild and asymptomatic cases [27]. Since there is no pertussis vaccination program in China after 18 months of age, the increase in Anti-PT-lgG antibodies after the age of 3 is mainly due to pertussis infection. Based on the Anti-PT-lgG decay rate [4], this study estimated an annual pertussis infection rate of 9321 (95% CI: 3336–16039) per 100,000 individuals aged 3 years and older. This estimated infection rate is much higher than the reported incidence of pertussis [6,12], which may be attributed to atypical or asymptomatic presentations, limited adoption of rapid diagnostic technologies, and variations in physician awareness and emphasis on diagnosis. Similar situations were observed in other regions, with estimated pertussis infection rates as follows: 7290/ 100,000 in Chongqing [19], 4573/100,000 in Ningbo [7], 9395/100,000 in Guangdong [25], 18080/100,000 in Jiangsu [17], and 6600/100,000 in the Netherlands [4]. Meanwhile, this study revealed a high susceptibility rate of 40.49% for individuals with Anti-PT-lgG <5 IU/mL,

consistent with findings from multiple studies [17,18,28]. This indicates a coexistence of high infection rates and susceptibility in the population.

Vaccination [29,30] and age [4] are crucial factors influencing Anti-PT-lgG levels. Pertussis is a vaccine-preventable disease, and since the introduction of the vaccine, the incidence of pertussis has consistently decreased, maintaining at very low levels of prevalence [20]. This study revealed that infants who did not receive the vaccine had a GMC of only 9.18 IU/mL, much lower than the Anti-PT-lgG levels in individuals who received 2 or more doses, with a susceptibility rate as high as 44.44%. This suggests that newborns in our study have very low levels of Anti-PT-IgG in their bodies before receiving the pertussis vaccine, making them a high-risk group susceptible to infection when exposed to a source of infection. In fact, pertussis toxin antibodies are readily transferred across the placenta with high efficiency, resulting in infant serum antibody levels that are essentially equivalent to those of the mother [31,32]. Since our country does not have a strategy for vaccinating adults or pregnant women against pertussis, it is very likely that maternal levels of pertussis toxin antibodies are also low. As our study found, pertussis toxin antibody levels in women of childbearing age are generally low. Studies have shown that maternal vaccination against pertussis during pregnancy results in a GMC of Anti-PT-lgG >30 IU/mL in newborns at birth [33].The results of this study showed a significant increase in Anti-PT-lgG levels with the number of vaccine doses administered. Research indicates that antibodies induced by pertussis vaccines begin to decline 3–5 years after the last vaccine dose, decreasing to only 0%-20% within 10 years [30]. This study's findings also demonstrate a rapid decline in Anti-PT-lgG levels over time after full vaccination no matter the different kind of pertussis vaccines, with the 2.43–5.59 year group having Anti-PT-lgG concentrations only 28.64% of the <0.7 year group. Existing study [34] demonstrates that regardless of which pertussis vaccine is administered, antibody levels decline over time. This suggests a rapid decay of vaccine-induced antibodies, with a relatively short duration of antibody maintenance. By the age of 10, Anti-PT-lgG levels in different age groups have converged to lower levels, with susceptibility rates exceeding 40% in all groups. The susceptibility rate in the 20–29 year group reaches 67%, estimating an annual infection rate as high as 34362.58/ 100,000, followed by the 40–49 year and 5–9 year groups, representing the primary population of school-age children and those caring for them. Some studies have shown that susceptibility increases 5–10 years after vaccination [35]. Therefore, many countries introduce a booster dose at school entry (4–6 years old) to extend individual antibody protection, effectively reducing the infection rate in this age group [36].

In this study, the susceptibility rate of women of childbearing age (15–49 years old) is as high as 54.88%, with a GMC of only 7.71 IU/mL, indicating a low level of Anti-PT-lgG in maternal antibodies. Chen et al.'s study [28] found that the proportion of women of childbearing age with Anti-PT-lgG <5 IU/mL was 66.66%, 65.99%, and 70.24% in the years 2018–2020, showing an increasing trend in susceptibility. The median Anti-PT-lgG in 2020 was significantly lower than the pre-pandemic levels in 2018 and 2019. Consistent with the results of this study, women of childbearing age have low levels of Anti-PT-lgG and high susceptibility. Maintaining a high level of Anti-PT-lgG in women of childbearing age helps boost maternal antibody levels, protecting infants who have not yet received vaccinations from infection [37]. Therefore, there are currently recommendations for pregnant women to receive pertussis vaccination during pregnancy [38]. Australia actively recommends pregnant women to receive Tdap vaccination during the last three months of pregnancy [18], and the United States has a policy for women of childbearing age to receive 1 dose of Tdap [39]. However, there is currently no specific policy regarding pertussis vaccination for pregnant women in China. In addition, adults caring for infants also have relatively low levels of Anti-PT-lgG. In this study, the GMC of Anti-PT-lgG in the population aged 20 and above is below 10 IU/mL. Adult

infections often exhibit atypical or asymptomatic symptoms [40], making them challenging to detect in a timely manner. The misdiagnosis rate can be as high as 94.69% [41], but they still remain infectious and represent important sources of infection for susceptible children [40]. Therefore, countries like the United States, France, Germany, and Australia have adopted the "cocooning strategy," recommending pertussis vaccination for close contacts of newborns [29].

This study has several limitations. Firstly, the lack of standardized values for pertussis ELISA testing poses challenges in interpreting results, as judgments depend on the cutoff values set by the test kits, impacting comparisons with other studies. There are no established standards for positive levels of Anti-PT-lgG, making it challenging to evaluate the positive rate of Anti-PT-IgG of different population. Although diverse, the overall conclusions of various studies are consistent. Secondly, the study population is exclusively from Huzhou, and the number of different age group samples was small. It is highly likely that the acute infection rate being 0 in the 30–39 age group is due to insufficient sample size in the subgroup. The age structure of the study samples is inconsistent with that of the population, which may limit the generalizability of the findings. Nevertheless, the results align with existing research, providing valuable insights into the pertussis epidemiology in Huzhou. Lastly, the study focused on Anti-PT-lgG detection, which may not fully represent pertussis antibody levels. However, research indicates that Anti-PT-lgG is highly specific and representative of pertussis antibodies, to a large extent reflecting pertussis antibody levels. There was no comparison of PT-IgG concentrations after different types of pertussis vaccine administration.

## Conclusions

In summary, the seroepidemiological study suggests a relatively high actual pertussis infection rate, rapid decay of vaccine-induced antibodies, and generally low levels of Anti-PT-lgG and heightened susceptibility in the population of Huzhou. It is essential to strengthen pertussis monitoring and promptly refine pertussis vaccination strategies.

## Supporting information

**S1 File.**
(XLSX)

## Author Contributions

**Conceptualization:** Yan Liu, Jianyong Shen.

**Data curation:** Chao Zhang, Xiaofu Luo.

**Formal analysis:** Yan Liu.

**Investigation:** Yuda Wang, Xiaofu Luo.

**Methodology:** Guangtao Liu.

**Writing – original draft:** Yan Liu, Chao Zhang.

**Writing – review & editing:** Zizhe Zhang, Jianyong Shen.

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
