## [Decision Letter · Decision Letter 0]

29 Feb 2024

PONE-D-23-43849Seroepidemiology of pertussis in Huzhou: A population-based, cross-sectional studyPLOS ONE

Dear Dr. liu,

Thank you for submitting your manuscript to PLOS ONE. After careful consideration, we feel that it has merit but does not fully meet PLOS ONE’s publication criteria as it currently stands. Therefore, we invite you to submit a revised version of the manuscript that addresses the points raised during the review process, particularly concerning the sample size and sampling methods adopted in this seroepidemiological study.

We look forward to receiving your revised manuscript.

Kind regards,

Daniela Flavia Hozbor

Academic Editor

PLOS ONE

Journal Requirements:

Reviewers' comments:

Reviewer's Responses to Questions

**Comments to the Author**

1. Is the manuscript technically sound, and do the data support the conclusions?

Reviewer #1: Yes

Reviewer #2: Partly

Reviewer #3: Yes

2. Has the statistical analysis been performed appropriately and rigorously? 

Reviewer #1: Yes

Reviewer #2: I Don't Know

Reviewer #3: Yes

3. Have the authors made all data underlying the findings in their manuscript fully available?

Reviewer #1: Yes

Reviewer #2: No

Reviewer #3: Yes

4. Is the manuscript presented in an intelligible fashion and written in standard English?

Reviewer #1: Yes

Reviewer #2: No

Reviewer #3: Yes

5. Review Comments to the Author

Reviewer #1: The authors in the study assess the level of pertussis infection in the Huzhou (Cina) population, to evaluate the opportunity to adjust pertussis prevention and control strategies. some points need clarification.

It is not indicated which vaccine preparation was used in the vaccinated individuals and whether all vaccinated people were given the same preparation. In fact the response to PT could be different in relation to different vaccine preparation.

Another point to consider is the execution of the tests if they have been carried out by the same laboratory. Can you specify that?

Check that all microbial names are italicized.

Reviewer #2: The study was conducted in Huzhou City, China, in 2023. Serum samples were collected from 1015 permanent residents and tested for pertussis toxin IgG antibodies (Anti-PT-IgG) using ELISA. Although pertussis, a vaccine-preventable disease, remains an important health problem among children, I have several concerns about the sampling method and sample size adopted in this cross-sectional survey.

1) It is not clear how many urban districts and counties there are in the city of Huzhou. Why was only one urban district and county selected? How representative are the estimates of the city of Huzhou?

2) After randomly selecting the urban district and county, how did the authors select the participants? Did they visit households? How were households sampled? How many individuals were enrolled from each household? The methods section needs to elaborate on the sampling procedure.

3) Although the study had nine age bands, the number of samples is very small to arrive at any meaningful inference.

4) One sample size for such a large city is not sufficient. Authors should have divided the city into different zones/regions and should have calculated the sample size at region/zonal level. This would have enabled authors to look for any spatial variations within the city.

5) How the number of participants per age group was fixed? Does it represent the age structure of the city population?

6) I suggest the authors use the STROBE checklist for cross-sectional studies

Reviewer #3: Line 18: Consider rewording the sentence

Line 34: Consider representing the infection rate as a whole number while still including the 95% CI to the hundredth place

Line 56: What age groups are used to signify infants, adolescents, and adults? Does Adults include +65 years of age?

Line 61: Please explain what you mean that pertussis toxin is a protective antigen. Yes, it is unique to bordetella pertussis thus specific but I don't understand what you mean it is protective antigen when it is an exotoxin that inhbiits neutrophil and macrophage recruitment in the early stages of infection.

Line 85: Consider different phrasing instead of 'so on'

Line 116: Maternal antibodies typically clear in the first 6 months, does the analysis change if you looked at greater than or equal to 2 years? Is the 3 year criteria based more on the 4th dose at 18 months?

Line 141: With the schedule being 3,4,5 months is it known if any of the susceptible cases where unvaccinated or only received 1 dose of vaccine?

Table 1: Do the authors have any conclusions as to why the Age 30-39Y group has no acute infections while all other groups have acute infections

Table 1: There is a type in the Table 40-4Y instead of 40-49Y

Table 2: It is not clear if any of the 0-3 months (Total 97 enrolled) are included in this table. Would that group represent a large portion of the zero vaccines group?

Line 235: I don't think this is a correct assumption. There is no mention of matched infant-mother sera to establish low transfer between the mother and infant. Maternal antibody transfer is primarily based on the mother's antibody level for pertussis and there is active transport during the third trimester of pregnancy. Additionally protective suggests functional antibody which is not what the ELISA measures. You go on to explain the benefits of maternal immunization in other countries correctly but the initial statement of low transfer is not proven in this paper. Please refine the statement in Line 235

Line 276: Are you referring to correlates of protection or fuctionally protective antibodies?

6. PLOS authors have the option to publish the peer review history of their article (what does this mean?). If published, this will include your full peer review and any attached files.

Reviewer #1: No

Reviewer #2: No

Reviewer #3: No

---

## [Author Response · Author response to Decision Letter 0]

5 Apr 2024

We thank the editor and reviewers for their constructive inputs to our paper. We have addressed all the comments. Please find our point-to-point responses in the "response to reviewers".

---

## [Decision Letter · Decision Letter 1]

26 Apr 2024

Seroepidemiology of pertussis in Huzhou: A population-based, cross-sectional study

PONE-D-23-43849R1

Dear Dr. Yan liu,

We’re pleased to inform you that your manuscript has been judged scientifically suitable for publication and will be formally accepted for publication once it meets all outstanding technical requirements.

Kind regards,

Daniela Flavia Hozbor

Academic Editor

PLOS ONE

Additional Editor Comments (optional):

Reviewers' comments:

Reviewer's Responses to Questions

**Comments to the Author**

1. If the authors have adequately addressed your comments raised in a previous round of review and you feel that this manuscript is now acceptable for publication, you may indicate that here to bypass the “Comments to the Author” section, enter your conflict of interest statement in the “Confidential to Editor” section, and submit your "Accept" recommendation.

Reviewer #2: All comments have been addressed

Reviewer #3: All comments have been addressed

2. Is the manuscript technically sound, and do the data support the conclusions?

Reviewer #2: Yes

Reviewer #3: Yes

3. Has the statistical analysis been performed appropriately and rigorously? 

Reviewer #2: I Don't Know

Reviewer #3: Yes

4. Have the authors made all data underlying the findings in their manuscript fully available?

Reviewer #2: Yes

Reviewer #3: Yes

5. Is the manuscript presented in an intelligible fashion and written in standard English?

Reviewer #2: (No Response)

Reviewer #3: Yes

6. Review Comments to the Author

Reviewer #2: (No Response)

Reviewer #3: (No Response)

7. PLOS authors have the option to publish the peer review history of their article (what does this mean?). If published, this will include your full peer review and any attached files.

Reviewer #2: No

Reviewer #3: No

---

## [Editor Report · Acceptance letter]

30 Apr 2024

PONE-D-23-43849R1 

PLOS ONE

Dear Dr. Liu, 

I'm pleased to inform you that your manuscript has been deemed suitable for publication in PLOS ONE. Congratulations! Your manuscript is now being handed over to our production team.

Kind regards, 

on behalf of

Dr. Daniela Flavia Hozbor 

Academic Editor

PLOS ONE